# Advances in Adjuvanted Influenza Vaccines

**DOI:** 10.3390/vaccines11081391

**Published:** 2023-08-21

**Authors:** Shintaro Shichinohe, Tokiko Watanabe

**Affiliations:** 1Department of Molecular Virology, Research Institute for Microbial Diseases, Osaka University, Osaka 565-0871, Japan; 2Center for Infectious Disease and Education and Research (CiDER), Osaka University, Osaka 565-0871, Japan; 3Center for Advanced Modalities and DDS (CAMaD), Osaka University, Osaka 565-0871, Japan

**Keywords:** influenza vaccine, seasonal influenza, pandemic influenza, adjuvant, adverse effects, dose sparing, broad immune response

## Abstract

The numerous influenza infections that occur every year present a major public health problem. Influenza vaccines are important for the prevention of the disease; however, their effectiveness against infection can be suboptimal. Particularly in the elderly, immune induction can be insufficient, and the vaccine efficacy against infection is usually lower than that in young adults. Vaccine efficacy can be improved by the addition of adjuvants, and an influenza vaccine with an oil-in-water adjuvant MF59, FLUAD, has been recently licensed in the United States and other countries for persons aged 65 years and older. Although the adverse effects of adjuvanted vaccines have been a concern, many adverse effects of currently approved adjuvanted influenza vaccines are mild and acceptable, given the overriding benefits of the vaccine. Since sufficient immunity can be induced with a small amount of vaccine antigen in the presence of an adjuvant, adjuvanted vaccines promote dose sparing and the prompt preparation of vaccines for pandemic influenza. Adjuvants not only enhance the immune response to antigens but can also be effective against antigenically different viruses. In this narrative review, we provide an overview of influenza vaccines, both past and present, before presenting a discussion of adjuvanted influenza vaccines and their future.

## 1. Introduction

Influenza is an acute, highly contagious respiratory disease caused by influenza A and B viruses with symptoms that include congestion, fever, and body ache. Influenza epidemics are caused every winter by several different circulating strains of seasonal influenza viruses (i.e., A/H1N1, A/H3N2, B/Yamagata, and B/Victoria). Influenza places a substantial burden on our health; the World Health Organization (WHO) estimates that influenza causes 3–5 million cases of severe illnesses that result in 290,000–650,000 deaths each year globally [1,2]. In particular, people 65 years and older account for 70–85% of flu-related deaths and 50–70% of flu-related hospitalizations [3], although the influenza activity has decreased in recent seasons (i.e., 2020–2021 and 2021–2022) due to the COVID-19 pandemic. Moreover, occasional pandemics are caused by novel strains of influenza A viruses and result in considerable morbidity and mortality globally. The most recent pandemic was caused by A(H1N1)pdm09, which was transmitted from pigs to humans in 2009 and currently causes annual epidemics as seasonal A/H1N1. Children, elderly people, and adults with chronic health conditions (e.g., asthma, diabetes, chronic kidney disease, or heart disease) are at a high risk for influenza-related complications [4]. Additionally, avian influenza A viruses, such as the H5N1 and H7N9 subtypes, have been reported to sporadically infect humans, resulting in 458 deaths and 616 deaths, respectively, as of 24 April 2023, with limited evidence of sustained human-to-human transmission [5].

Various vaccines and antiviral drugs are used to prevent and treat influenza; however, the effectiveness of the current influenza vaccine against infection can be suboptimal [6]. One of the reasons is that the current vaccines for seasonal influenza are manufactured several months before the next influenza season begins, and therefore, the antigenicity of the vaccine strains may not be perfectly matched with that of the dominant strains circulating in the next season. Moreover, elderly people generally show lower immune responses and vaccine effectiveness against infection. Their mild immunogenicity can be improved by the addition of adjuvants. An ideal adjuvant should increase vaccine efficacy and have a strong safety profile [7]. In this review, we present an overview of the licensed influenza vaccines and the current use of adjuvants for them. In addition, we discuss recent challenges in the development of adjuvanted influenza vaccines.

## 2. Current Licensed Influenza Vaccines

Annual vaccination is the most effective way to prevent seasonal influenza infection [8]. Currently, three types of licensed seasonal influenza vaccines are available for human use: inactivated, live attenuated, and recombinant hemagglutinin (HA) vaccines (Table 1). These vaccines are composed of three or four different types of influenza viruses, which are updated annually by the WHO to reflect the most recent circulating strains [9]. Current influenza vaccines are generally trivalent or quadrivalent, containing two influenza A strains (i.e., A/H1N1 and A/H3N2) and one or two influenza B viruses that are predicted to circulate in the upcoming season. Vaccine effectiveness against virus infections varies from season to season depending on the match or mismatch between the vaccine and circulating strains [10,11]. Various types of influenza vaccines have been developed so far, and the approved vaccines are described below.

### 2.1. Inactivated Vaccines

There are three types of inactivated influenza vaccines: whole virus particle vaccines, split virus vaccines, and subunit vaccines. The inactivated whole virus particle vaccine, developed in the 1940s, consists of virus particles that lack infectivity but maintain the shape of viral particles, including viral genomic RNAs and all viral structural proteins. The inactivated whole virus particle vaccine can induce humoral and cellular immune responses effectively [12,13]. Whole virus particle inactivated vaccines retain the ability to enter target cells but do not propagate, and genomic RNA in viral particles stimulates Toll-like receptor (TLR) 7 [13]. However, there were concerns about pyrogenicity and adverse reactions [14], presumably due to the lipid components of the virus particles, and, therefore, this vaccine was replaced with the split virus vaccine, which is prepared by treating the purified virus with ether or detergents to remove the viral envelope [15]. Subunit vaccines include mainly HA and neuraminidase proteins. Split and subunit vaccines induce immunity in people who have previously been infected with the influenza virus; however, they do not induce sufficient immunity in infants who have never been infected. This is because split vaccines lose much of the viral single-stranded (ss) RNA and have reduced immunogenicity due to a lack of signaling to innate immune receptors [13]. Thus, activation of innate immunity is important for vaccines to induce robust immunity.

In the case of inactivated whole virus particle vaccines, viral genomic RNA in viral particles is taken up by dendritic cells (DCs) along with other viral components and then stimulates the pattern recognition receptor signaling cascade, leading to the maturation of the DCs [16]. In contrast, in the case of split or subunit vaccines, HA protein is incorporated into the DCs, but viral RNA is not effectively incorporated into the DCs, resulting in inadequate DC maturation [16]. Some antigens are directed to endolysosomes, where they access the MHC I antigen processing pathway, called the “cross-presentation pathway”, where antigen epitopes are loaded onto MHC I molecules and activate CD8+ cytotoxic T cells. Whole virus particle vaccines can present all epitopes of the structural viral proteins on the MHC molecules, resulting in the induction of stronger and broader humoral and cellular adaptive immunity in vaccinated individuals compared to split or subunit vaccines [16].

### 2.2. Live Attenuated Vaccines

Live attenuated influenza vaccines (LAIV) are based on cold-adapted, temperature-sensitive, and attenuated variants that are produced by consecutive passages in embryonated chicken eggs at low temperatures. For example, the virus in FluMist^®^ has surface glycoproteins that come from circulating viruses, whereas its internal genes come from cold-adapted master donor viruses [17]. Therefore, upon intranasal vaccination with these vaccines, their replication is limited to the upper respiratory tract, where the temperature is lower than normal body temperature. Similar to natural infection, live attenuated vaccines induce mucosal IgA responses in the upper respiratory tract and cross-reactive T-cell responses. Double-stranded (ds) viral RNA, which is produced during virus replication, is recognized by endosomal innate immune receptor TLR 3 [18,19] and retinoic acid inducible gene (RIG)-I, leading to induction of interferon-mediated antiviral responses and proinflammatory cytokine responses.

Nasal spray vaccines such as FluMist^®^ are live attenuated influenza vaccines that are administered to healthy people between the ages of 2 and 49 who are not pregnant and have no history of medical issues [20]. The nasal spray vaccine used in recent seasonal influenza seasons contains A/H1N1 and A/H3N2 and two influenza B viruses. Since the 2009 pandemic, the effectiveness of this vaccine against A/H1N1 infection has decreased, and vaccination with it was temporarily not recommended in the U.S. The decline in vaccine effectiveness is thought to be most likely due to a decrease in the replicative fitness of the LAIV vaccine strain against A(H1N1)pdm09 infection, resulting in reduced protective immunity [21]. After changing to a new vaccine strain for the 2017–2018 season, vaccinations have restarted and have been reported to be as effective as the single-dose, egg-based, inactivated whole virus particle vaccine for children [22].

### 2.3. Recombinant HA Vaccines

Recombinant HA vaccines contain purified HA that is produced in insect cells by using a baculovirus expression system. Flublok^®^ quadrivalent vaccine is a recombinant influenza vaccine that was approved for people 18 years and older in the U.S. and EU in 2017 and 2020, respectively. This vaccine has been available in the U.S. and certain countries in the EU since the 2020–2021 influenza season. Individuals vaccinated with Flublok^®^ had a 30% lower probability of influenza-like illness than those vaccinated with a standard-dose, egg-grown, quadrivalent, and inactivated influenza vaccine, most likely because the recombinant HA vaccine contained three times the amount of HA protein [23]. During Flublok^®^ production, seed virus and chicken eggs are not required, resulting in faster production of recombinant vaccine than for egg- or cell-based vaccines because egg or cell adaptation of seed viruses for viral growth is not a limitation.

### 2.4. Effectiveness of Licensed Seasonal Influenza Vaccines

The CDC has reported on the vaccine effectiveness (VE) of influenza vaccines from the 2004–2022 influenza seasons, although VE estimates for the 2020–2021 season are not available due to low influenza virus circulation during that season [24]. Although participants who tested positive for severe acute respiratory syndrome coronavirus 2 (SARS-CoV-2) were excluded among controls, the 2021–2022 VE against A/H1N1 and A/H3N2 in Europe was 79% and 36% in all ages, respectively [25].

Influenza viruses are constantly changing due to antigenic drift, making it difficult to predict the virus that will prevail in the next season [26]. If the antigenicity of the vaccine strain and the epidemic strain does not match, the effectiveness against infection of the influenza vaccine is reduced. In particular, the VE of H3N2 influenza vaccines is lower than that of A (H1N1) pdm09 and influenza B vaccines, a phenomenon that is associated with advanced genetic variation. Recent seasonal H3N2 influenza viruses replicate poorly in eggs, requiring vaccine strains to be adapted by passaging in eggs to increase production efficiency. Furthermore, egg adaptation causes mutations in HA that lead to antigenic alteration of the H3N2 vaccine strain and poor immunological responses [27,28].

The WHO selects vaccine strains by considering the epidemic conditions around the world; however, there have been several cases where the antigenicity has not matched, resulting in the suboptimal effectiveness of the vaccines against infection [29]. High-dose vaccines (e.g., Fluzone High Dose) and vaccines with adjuvants (e.g., FLUAD^®^, MF59-adjuvanted influenza vaccine) have been developed as more effective vaccines. These vaccines show promise, having higher VE than conventional vaccines [30,31].

## 3. Adjuvants Used for Licensed Influenza Vaccines

One of the most effective strategies to improve vaccine efficacy against infection is the addition of adjuvants. Adjuvants are added to or administered with the vaccine antigen to enhance the specific immune response. In the case of influenza vaccines, adjuvants are effective in enhancing immunogenicity, preventing hospitalization of the elderly, and reducing the amount of viral antigen needed for protection and the number of vaccinations required. In addition, adjuvants can broaden reactivity to antigenic variants and are effective in combating antigenic mismatches between vaccine strains and circulating viruses [32,33]. Generally, it is known that adjuvants induce immune responses through mechanisms such as depot effects, induction of cytokine responses, recruitment of immune cells, promotion of antigen uptake and dendritic cell maturation, release of host dsDNA, activation of innate signaling pathways and inflammasomes, generation of Th1 and/or Th2 immune responses and follicular helper T cells [34,35]. A number of adjuvants have been approved for use in influenza vaccines, including alum, MF59, AS03, AF03, and virosome [36] (Table 2).

### 3.1. Alum

Aluminum salts (mainly alum), including aluminum hydroxide, aluminum phosphate, and aluminum potassium sulfate, are the most common adjuvants and have been used in vaccines since the 1930s [37,38]. Alum is thought to act by at least two independent mechanisms: (i) enhancing the immunogenicity of antigens by making them multivalent and facilitating their delivery to antigen-presenting cells, and (ii) promoting the local inflammatory environment leading to NOD-like receptor protein 3 (NLRP3)-mediated blood cell collection and dendric cell differentiation by a TLR-independent mechanism [39]. Aluminum salts are used in several licensed vaccines, including the Diphtheria, Pertussis, and Tetanus (DPT) vaccine. After the DPT vaccine with an aluminum adjuvant was successfully shown to elicit protective immunity in humans, an influenza vaccine combined with an aluminum adjuvant was developed; however, it did not show sufficient immune responses [40,41]. Despite the development of alum, research on influenza vaccine adjuvants did not make significant progress until the development of MF59.

The use of alum in vaccines will continue because of its well-known safety and cost-effectiveness. However, alum only induces humoral immunity, not cellular immunity [42]. Alum is also known to induce Th2-type immune responses characterized by the production of cytokines such as IL-4 and IL-5 [43]. These cytokines promote class switching of B cells to IgG1 and IgE isotypes. Thus, alum adjuvants often result in the production of IgG1 and, to a lesser extent, IgE antibodies. Different antibody isotypes may contribute to protection against influenza virus infection; IgG1 antibodies are often associated with alum adjuvant vaccines and provide neutralizing and opsonizing capacity, while IgG2a/c antibodies may contribute to antibody-dependent cellular cytotoxicity (ADCC) and complement activation. Other adjuvants, such as nucleotide analogs or lipids, might be required to elicit cellular immunity. Moreover, a desirable vaccine should be more immunogenic and less reactogenic. Therefore, a combination of approved adjuvants may be beneficial to develop a vaccine with higher immunogenicity, lower adverse effects, and lower cost.

### 3.2. MF59

MF59 is composed of an oil-in-water emulsion of squalene, emulsified with two surfactants (polysorbate 80 and sorbitan trioleate) [44,45]. The adjuvant effect of MF59 is thought to be due to the recruitment of immune cells to the injection site rather than a depot effect [46], and MF59 induces both Th1- and Th2-type immune responses and the release of some specific cytokines/chemokines such as IFN-γ, CCL2, CCL3, IL-8, and IL-5 [47]. MF59 is used in the seasonal influenza vaccine, FLUAD, and enhances immune responses even in the elderly [48,49,50]. FLUAD was first approved in Italy in 1997 and subsequently in the U.S. and Europe in 2015 for use among people 65 years and older [51]. Moreover, FLUAD quadrivalent influenza vaccine has also been approved for this elderly population and was available during the 2020–2021 season in the U.S. MF59 adjuvanted seasonal influenza vaccine elicits stronger immune responses in elderly adults and children compared with unadjuvanted vaccines [3,51].

Since continuous antigenic drift occurs frequently in influenza viruses, influenza epidemics occur every year. To combat antigenic variants, vaccines should induce broader antibody responses. Adjuvants in vaccines elicit cross-reactive and neutralizing antibodies and increase antibody titers [52,53,54]. Indeed, MF59 increased the diversity of the antibody epitope repertoire as well as the antibody binding affinity with H1N1pdm09 and H5N1 viruses [55,56], and thus, a combination of influenza vaccines and appropriate adjuvants may induce broader immune responses. Broader immune responses also have considerable manufacturing advantages, allowing for dose-sparing of vaccines and faster distribution in pandemics.

### 3.3. AS03

AS03 (GlaxoSmithKline (GSK)) is an adjuvant consisting of squalene, DL-α-tocopherol, and polysorbate 80 and is used as a vaccine for A(H1N1)pdm09 and H5N1 avian influenza viruses. AS03 triggers a transient NF-κB-dependent innate immune response, resulting in the production of cytokines and chemokines, including CCL2, CCL3, IL-6, CSF3 and CXCL1, at the injection site and in draining lymph nodes, which induces migration of immune cells [57]. AS03-adjuvanted A(H1N1)pdm09 vaccines induce higher antibody responses than non-adjuvanted vaccines in elderly people [57], despite a small number of antigens [58]. It differs from the oil-in-water adjuvants AS03 and MF59 in the phase inversion temperature emulsification process; however, the mechanism of action is equivalent [59]. AS03 promotes the development of T cells with increased antibody titers, but these tend to be mixed or preferentially biased toward Th2 responses [60]. AS03 adjuvant is also known to contribute to the mixed induction of different isotypes, IgG1 and IgG2a antibodies.

### 3.4. AF03

AF03 (Sanofi Pasteur) is an oil-in-water adjuvant used in the split-virion H1N1 pandemic influenza vaccine Humenza, but it has never been on the market [59]. Although AF03 has been reported to potentiate humoral and cellular (i.e., IFN-γ and IL-5) immune responses in animal models [61], the mechanisms are still largely unclear [36,62]. The A(H1N1)pdm09 vaccine induced a sufficient antibody response against influenza virus in primed and unprimed mice when administered with AF03 [63].

### 3.5. Virosomes

Virosomes, which have influenza antigens on their surface, mimic viruses and can be considered a kind of virus-like particle (VLP). By mimicking a virus, virosomes can take antigens into antigen-presenting cells and trafficking them within the lymph system. Virosome-based vaccines include Inflexal^®^ V for influenza [64]. Inflexal^®^ V was a trivalent virosome vaccine containing HA and NA from the WHO-recommended vaccine strains. After it was approved in Switzerland in 1997, it was introduced to markets all over the world but is now no longer available. It induces a strong immune response in both healthy and immunocompromised elders, adults, and children, similar to natural viral infections [65].

## 4. Adverse Effects Related to Adjuvanted Vaccines

Influenza vaccines are sufficiently effective considering the risks and benefits; however, side effects must also be considered. Safety concerns regarding the oil-in-water adjuvant AS03 were raised in the pandemic H1N1 influenza vaccine. Local and systemic adverse effects of this vaccine were reported, although the effects were not severe [66]. AS03-adjuvanted vaccines have been linked to an increased incidence of narcolepsy in children in some countries [66,67,68]. However, recent studies suggest that the trigger for narcolepsy may be viral protein(s) of the pandemic H1N1 influenza viruses, not the adjuvant AS03 [69]. Narcolepsy type 1 is caused by the autoimmune destruction of hypocretin neurons [70]. Localized CD4+ T cell responses recognizing the hypocretin peptide may have been stimulated by the viral antigen(s) of the pandemic H1N1 influenza virus due to the molecular mimicry between hypocretin and the viral antigen peptides, leading to cross-reactive autoimmune responses targeting hypocretin neurons [69,71,72].

The antibody responses of MF59 and alum in the recombinant split vaccine for the H5N1 avian influenza virus have been evaluated in clinical trials [41]. MF59 increased the neutralizing antibody response to viruses compared to alum. No serious adverse effects due to vaccination were observed. There was no difference in systemic reactions, such as fever, malaise, and headache, between the aluminum adjuvant and MF59 adjuvant groups, although local pain and tenderness were predominantly reported in 70% of the MF59-inoculated group. In addition, although no causal relationship between vaccines and adjuvants has been confirmed for adverse effects, it is important to continue to investigate any causal relationship to improve vaccine safety.

## 5. Progress in the Development of Adjuvants for Influenza Vaccines

### 5.1. Saponin

Although alum has been used in vaccines for a long time, this adjuvant induces humoral but not cellular immunity [38]. Saponin-based adjuvants, which are present in vaccines for veterinary use, induce both humoral and cellular immunity [73]. Although Quil A and its derivative QS-21 have been assessed as influenza vaccine adjuvants in humans, there are no clear advantages in terms of immunity induced by vaccines with or without these adjuvants [74,75]. A recent screen of compounds from food additives identified saponin as a good adjuvant candidate since it enhanced the immune response elicited by the split influenza vaccine in mice [76].

Following the emergence of the H7N9 avian influenza virus in 2013, China developed the first H7N9 vaccine in October of the same year to prepare for the possible spread of infection among humans. The following month, researchers from Novavax Inc. (Gaithersburg, MD, USA) announced that their clinical trials of an H7N9 vaccine were successful [77]. They reported the HI antibody responses of a saponin-based ISCOMATRIX adjuvant added to the H7N9 VLP vaccine, using a baculovirus expression system, with the HA and NA proteins from A/Anhui/1/2013 (H7N9) and the M1 protein from A/Indonesia/5/2005 (H5N1). Although the H7 virus is considered to elicit lower immunogenicity, the addition of a saponin-based adjuvant elicits higher immunogenicity than that induced by alum or no adjuvant [78]. Thus, like the results with the seasonal influenza vaccine, saponin-based adjuvants are also immunogenic when combined with the H7 virus vaccine.

Matrix M^TM^ is a saponin-based adjuvant that has been evaluated in Phase 1 clinical trials as an adjuvant for the virosomal H5N1 influenza vaccine [79]. The addition of Matrix M^TM^ to the H5N1 vaccine induced a sufficient antibody response against both homologous and heterologous H5N1 influenza virus [79,80]. Matrix M^TM^ adjuvanted quadrivalent nanoparticle vaccine for seasonal influenza was also evaluated in Phase 3 randomized controlled trials. This vaccine elicited higher humoral and cellular immunity in older adults aged at least 65 years old, compared with the licensed quadrivalent inactivated influenza vaccine [75]. Matrix M^TM^ also required less antigen to induce the immune response compared with that in the vaccine alone, which will be important for dose sparing.

### 5.2. TLR Agonists

There has been an increase in the number of adjuvant studies focusing on the agonists of innate immune receptors. Innate immune receptors such as TLRs are directly activated by adjuvants [18].

Poly I:C is a synthetic TLR3 agonist with a structure like dsRNA. Poly I:C caused safety concerns in Phase 1/2 trials in cancer patients [81]. To reduce adverse events, poly I:C derivatives, poly I:C12U (Ampligen, Rintatolimod), poly-ICLC (Hiltonol^®^), and Polyinosinic-Polycytidylic Acid Based Adjuvant (PIKA) were developed. Poly I:C12U is synthesized with mismatched uracil and guanine bases in the RNA chain. Poly I:C12U has been tested as an adjuvant for nasal influenza vaccines and has been reported to exhibit high mucosal immunogenicity [82]. It has also been reported that combining Rintatolimod with seasonal influenza vaccines induces cross-reactive IgA against other subtypes of influenza [83]. PIKA is also a stabilized dsRNA that has been shown to be effective as an adjuvant in vaccines against H1, H3, and H5 subtype influenza viruses in animal studies by subcutaneous injection and intranasal immunization [84,85].

The synthetic lipid A analog glucopyranosyl lipid A (GLA) is a TLR4 ligand that has been examined as an adjuvant. The HI antibody titers of a vaccine comprising plant-derived VLPs expressing the HA of the H5N1 virus and either an alum-based adjuvant or glucopyranosyl lipid adjuvant-stable emulsion (GLA-SE) have been assessed in a Phase 2 study [86]. The group immunized with GLA-SE had higher HI antibody titers than that immunized with the alum-based adjuvant and required less antigen.

Flagellin, a major component of bacterial flagella, is also a TLR5 ligand that has been examined as an adjuvant. Recombinant protein expressed by *E. coli* that fuses Salmonella typhimurium flagellin type 2 (STF2) and the globular head domain of H1HA induced high HI antibody titers in healthy adults and the elderly in a Phase 1/2 trial [87]. This type of quadrivalent vaccine against seasonal influenza has also been evaluated in a phase 1 trial and induced sufficient HI antibody titers with 2–3 micrograms of antigen (total protein) per component without significant safety concerns.

In the past decade, much research has focused on the development of universal influenza vaccines, with an emphasis on developing vaccine strategies that target conserved regions of influenza virus proteins, such as the HA stalk, NA, and M2 [88]. Adjuvanted universal influenza vaccines that induce cross-protection against heterologous strains are also being investigated. In a recent report, the TLR7 agonist DSP-0546LP was shown to induce Th1-biased immunity in mice, such as IFN-γ and IgG2c antibody production, and enhance cross-protection against viruses of the H1N1 and H3N2 subtypes with antigenic distance. This was shown to be mainly due to ADCC rather than cross-neutralizing antibodies [89]. It has also been reported that ADCC-induced cross-protection immunity is induced not only by TLR7 but also by TLR9 [90]. Further studies using TLR agonists as adjuvants are needed, but they show promise as adjuvants that induce high reactogenicity and cross-reactive immune responses.

## 6. Conclusions

There was a long gap between the discovery of alum as an adjuvant about 100 years ago and the development of MF59. With recent advances in our understanding of the development of innate immunity, various new adjuvants are being developed. Adjuvants are useful to enhance the effectiveness of vaccines against infections; however, a high degree of compatibility (good affinity) is required for the combination of vaccine and adjuvant to be effective. Adjuvants not only enhance the immune response but also expand the range of responses to antigenic variants. Therefore, adjuvanted vaccines seem to be suitable against influenza viruses in which antigenic drift frequently occurs. The addition of adjuvants also enables dose sparing of the antigen required to induce sufficient immunity, which will allow rapid distribution of vaccines during an influenza pandemic. Therefore, adding an adjuvant to the influenza vaccine has several benefits. However, although few side effects have been reported, efforts to further reduce the adverse effects of adjuvants should be ongoing. Further research on adjuvants will provide a deeper understanding of their function and facilitate the development of safer and more immunogenic adjuvants.

## Figures and Tables

**Table 1 vaccines-11-01391-t001:** FDA-approved seasonal influenza vaccines.

**Vaccine Type**	Adjuvant	Production Platform	Dose	Age Indication	Tradename (Manufacturer), Route
Split	None	Embryonated egg	45 µg (15 µg HA/strain)	≥6 mo	Afluria Quadrivalent (Seqirus), i.m./i.n. ^a^
	None	Embryonated egg	45 µg (15 µg HA/strain)	≥6 mo	Afluria Southern Hemisphere (Seqirus), i.m./i.n. ^a^
	None	Embryonated egg	45 µg (15 µg HA/strain)	≥3 yrs	Fluarix (Glaxosmithkline), i.m.
	None	Embryonated egg	45 µg (15 µg HA/strain)	≥6 mo	FluLaval (GlaxoSmithKline), i.m.
	None	Embryonated egg	60 µg (15 µg HA/strain)	≥6 mo	FluLaval Quadrivalent (GlaxoSmithKline), i.m.
	None	Embryonated egg	60 µg (15 µg HA/strain)	≥6 mo	Fluzone Quadrivalent (Sanofi Pasteur), i.m.
	None	Embryonated egg	180 µg (60 µg HA/strain)	≥65 yrs	Fluzone High Dose Quadrivalent (Sanofi Pasteur), i.m.
	None	Embryonated egg	36 µg (9 µg HA/strain)	18–64 yrs	Fluzone-Intradermal (Sanofi Pasteur), i.d.
Subunit	MF59	Embryonated egg	45 µg (15 µg HA/strain)	≥65 yrs	FLUAD (Seqirus), i.m.
	MF59	Embryonated egg	60 µg (15 µg HA/strain)	≥65 yrs	FLUAD Quadrivalent (Seqirus), i.m.
	None	Embryonated egg	45 µg (15 µg HA/strain)	≥4 yrs	Fluvirin (Seqirus), i.m.
	None	Embryonated egg	45 µg (15 µg HA/strain)	≥18 yrs	Agriflu (Seqirus), i.m.
	None	Madin-Darby canine kidney (MDCK) cell	60 µg (15 µg HA/strain)	≥6 mo	Flucelvax Quadrivalent (Seqirus), i.m.
Live Attenuated	None	Embryonated egg	10^6.5–7.5^ FFU ^b^ virus/strain	2–49 yrs	FluMist (AstraZeneca), i.n.
Recombinant	None	Baculovirus/Insect cell	135 µg (45 µg HA/strain)	≥18 yrs	FluBlok (Sanofi Pasteur), i.m.
			180 µg (45 µg HA/strain)	≥18 yrs	FluBlok Quadrivalent (Sanofi Pasteur), i.m.

^a^ PharmaJet Stratis jet injector can be used for persons aged 18 through 64 years only. ^b^ FFU, fluorescent focus units.

**Table 2 vaccines-11-01391-t002:** Adjuvants for licensed-influenza vaccines.

Adjuvant Name	Adjuvant Type	Components	Vaccines for
Alum	Aluminum salts	KAI(SO_4_)_2_, Aluminum phosphate, Aluminum hydroxide	Pre-pandemic influenza
MF59	Oil-in-water emulsion	Squalene, Tween 80, Sorbitan trioleate	Seasonal influenza, Pandemic influenza, Pre-pandemic influenza
AS03	Oil-in-water emulsion	Squalene, Tween 80, α-tocopherol	Pandemic influenza, Pre-pandemic influenza
AF03	Oil-in-water emulsion	Squalene, Polyoxyethylene cetostearyl ether, Mannitol, Sorbitanoleate	Pandemic influenza
Virosomes	Liposomes	Lipids, Hemagglutinin	Seasonal influenza

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
