# Peer review of "Advances in Adjuvanted Influenza Vaccines"

_vaccines, 2023, doi:10.3390/vaccines11081391_

Round 1
Reviewer 1 Report
This MS attempted to provide a concise overview on the advances of adjuvanted influenza vaccines, mainly involving licensed adjuvanted vaccines. Although lots of information has been mentioned, some of them is inaccurate or needs to be updated. Moreover, no advance in adjuvanted universal influenza vaccine was provided, which is the focus of attention in the field.
Key points
1. Line 14, it is better to provide the name of the approved influenza vaccines and the used adjuvant.
2. Table 1, it is not accurate to list FLUAD as “whole virus particle” vaccine. As indicated in the Package Insert of FLUAD Quadrivalent (2022), “The inactivated virus is concentrated and purified by zonal centrifugation. The surface antigens, hemagglutinin and neuraminidase, are obtained from the influenza virus particle by further centrifugation in the presence of cetyltrimethylammonium bromide (CTAB). The antigen preparation is further purified.” In fact, protein antigen in FLUAD Quadrivalent mainly consists of HA and NA as reported in recent publication (Fig8D, https://doi.org/10.1016/j.ijpharm.2022.122021).
3. Table 1, the information on Flucelvax should be updated. Currently, the target population of Flucelvax Quadrivalent has been divided into two groups: 6 months-8 years of age and 9 years of age-older (https://www.fda.gov/vaccines-blood-biologics/vaccines/flucelvax-quadrivalent).
4. Lines 190-191, is there any evidence to support the claim “A combination of influenza vaccines and appropriate adjuvants could be used for several years without the requirement for annually changing the vaccine strains.”? If yes, please provide the reference.
5. Lines 206-210, the claims on AF03’s mechanism of action seems contradictory.
6. Lines 217, inflexal V was discontinued in 2011 after Crucell joined Johnson & Johnson. Thus, it is not available in the market now.
7. Lines 256-258, the cited reference 73 only mentioned ISCOMATRIX, MF59, AS03 and aluminium adjuvants applied in H7N9 vaccines, but not oil-based adjuvants.
Author Response
[Reviewer #1]:
Comments and Suggestions for Authors
This MS attempted to provide a concise overview on the advances of adjuvanted influenza vaccines, mainly involving licensed adjuvanted vaccines. Although lots of information has been mentioned, some of them is inaccurate or needs to be updated. Moreover, no advance in adjuvanted universal influenza vaccine was provided, which is the focus of attention in the field.
In response to the reviewer’s comment, we have corrected and updated various statements in the revised manuscript as described below. We have also added information on adjuvanted universal influenza vaccine, as the reviewer suggested (page 10, lines 328–337).
Key points
- Line 14, it is better to provide the name of the approved influenza vaccines and the used adjuvant.
In response to the reviewer’s comment, we have added the name of the licensed influenza vaccine (FLUAD) with MF59 adjuvant (page 1, lines 14–15).
- Table 1, it is not accurate to list FLUAD as “whole virus particle” vaccine. As indicated in the Package Insert of FLUAD Quadrivalent (2022), “The inactivated virus is concentrated and purified by zonal centrifugation. The surface antigens, hemagglutinin and neuraminidase, are obtained from the influenza virus particle by further centrifugation in the presence of cetyltrimethylammonium bromide (CTAB). The antigen preparation is further purified.” In fact, protein antigen in FLUAD Quadrivalent mainly consists of HA and NA as reported in recent publication (Fig8D, https://doi.org/10.1016/j.ijpharm.2022.122021).
We have corrected the vaccine type of FLUAD to “subunit vaccine” in Table 1.
- Table 1, the information on Flucelvax should be updated. Currently, the target population of Flucelvax Quadrivalent has been divided into two groups: 6 months-8 years of age and 9 years of age-older (https://www.fda.gov/vaccines-blood-biologics/vaccines/flucelvax-quadrivalent).
As requested, we have updated the information on Flucelvax in Table 1.
- Lines 190-191, is there any evidence to support the claim “A combination of influenza vaccines and appropriate adjuvants could be used for several years without the requirement for annually changing the vaccine strains.”? If yes, please provide the reference.
Since there is no evidence to support this claim, we have modified the sentences in the revised manuscript (page 7, lines 210-213).
- Lines 206-210, the claims on AF03’s mechanism of action seems contradictory.
According to previously published information (European Medicines Agency, 2010), AF03 potentiates the IL-5 and IFN-γ immune responses, and therefore, we would like to leave the text as is with a minor modification (page 8, lines 232-234).
- Lines 217, inflexal V was discontinued in 2011 after Crucell joined Johnson & Johnson. Thus, it is not available in the market now.
We now state that Inflexal® V is not currently marketed (page 8, lines 241-246).
- Lines 256-258, the cited reference 73 only mentioned ISCOMATRIX, MF59, AS03 and aluminium adjuvants applied in H7N9 vaccines, but not oil-based adjuvants.
We have deleted the statement about oil-based adjuvants from the revised manuscript (page 9, line 287 and 289).

Reviewer 2 Report
The article “Advances in adjuvanted influenza vaccines” is a straightforward compilation of the different FDA approved influenza vaccines and the current and new adjuvants used in these vaccines. It will be useful to those who need to know the current state of the field. The writing does not go into much detail of the immunological mechanisms behind adjuvant action and the differences between the individual vaccines. A more in-depth discussion of the following points would make the review more useful.
1) In Section 2.1, the author should discuss the difference between the inactivated vaccine and the subunit vaccines in terms of the necessity for cross-presentation in the generation of cytotoxic T cells (CD8+). The differences are ascribed solely to differences in innate signaling rather than including differences in antigen presentation.
2) In Section 3.1, the specific cytokine responses with alum are not discussed. The effect of specific cytokines on antibody isotype and function should be discussed in the context of alum and with regard to other adjuvants. Is IgG isotype important in protection against influenza?
3) In Section 3.3, cytokines are mentioned generically, but the differences in specific cytokines produced between the different adjuvants should be explicitly mentioned. Also, the resulting effect on antibody isotypes should be mentioned if known.
A few grammatical corrections:
a. Line 207-208 “response against influenza in primed”
b. Line 218 immune response “in” both healthy
c. Line 242 “alum has been used” and “adjuvant induces humoral”
Author Response
[Reviewer#2]:
The article “Advances in adjuvanted influenza vaccines” is a straightforward compilation of the different FDA approved influenza vaccines and the current and new adjuvants used in these vaccines. It will be useful to those who need to know the current state of the field. The writing does not go into much detail of the immunological mechanisms behind adjuvant action and the differences between the individual vaccines. A more in-depth discussion of the following points would make the review more useful.
1) In Section 2.1, the author should discuss the difference between the inactivated vaccine and the subunit vaccines in terms of the necessity for cross-presentation in the generation of cytotoxic T cells (CD8+). The differences are ascribed solely to differences in innate signaling rather than including differences in antigen presentation.
In the case of inactivated whole virus particle vaccines, viral genomic RNA in viral particles is taken up by dendritic cells along with other viral components and then stimulates the pattern recognition receptor signaling cascade, leading to the maturation of dendritic cells (DCs) (Shingai et al., Vaccine, 2021). In contrast, in the case of split and subunit vaccines, HA and/or NA proteins are incorporated into the DCs, but viral RNA is not effectively incorporated into the DCs, resulting in inadequate DC maturation. Some antigens are directed to endolysosomes, where they access the MHC I antigen processing pathway, called the "cross-presentation pathway" where antigen epitopes are loaded onto MHC I molecules and activate CD8+ cytotoxic T cells. Whole virus particle vaccines can present all epitopes of the structural viral proteins on the MHC molecules, resulting in induction of stronger and broader humoral and cellular adaptive immunity in vaccinated individuals compared to split or subunit vaccines. We have added this information to the revised manuscript (pages 2-3, lines 87-98).
2) In Section 3.1, the specific cytokine responses with alum are not discussed. The effect of specific cytokines on antibody isotype and function should be discussed in the context of alum and with regard to other adjuvants.
In response to the reviewer’s comment, we have added some sentences about the cytokine responses and the IgG isotypes that are specific for alum (page 7, lines 183-190). We have also added information about responses of cytokines/chemokines specific to different adjuvants (page 7, lines 183-190; page 7, lines 198-200, lines 220-221, and lines 227-229; page 8, lines 232-234).
Is IgG isotype important in protection against influenza?
Different antibody isotypes may contribute to protection against influenza virus infection; IgG1 antibodies are often associated with alum adjuvant vaccines and provide neutralizing and opsonizing capacity, whereas IgG2a/c antibodies may contribute to antibody-dependent cellular cytotoxicity (ADCC) and complement activation. We have included this information in the revised manuscript (page 7, lines 186-190).
3) In Section 3.3, cytokines are mentioned generically, but the differences in specific cytokines produced between the different adjuvants should be explicitly mentioned. Also, the resulting effect on antibody isotypes should be mentioned if known.
In response to the reviewer’s comment, we have added information about the differences in specific cytokines/chemokines, together with the resulting antibody isotypes (if known), produced in different adjuvants (page 7, lines 183-190; page 7, lines 198-200, lines 220-221, and lines 227-229; page 8, lines 232-234).
Comments on the Quality of English Language
A few grammatical corrections:
- Line 207-208 “response against influenza in primed”
We have corrected the sentence (page 8, lines 234-236).
- Line 218 immune response “in” both healthy
We have corrected the sentence (page 8, lines 244-246).
- Line 242 “alum has been used” and “adjuvant induces humoral”
We have corrected the sentence (page 9, lines 272-273).

Reviewer 3 Report
Advances in adjuvanted influenza vaccines
The manuscript submitted by Shichinohe and Watanabe presents a narrative review of influenza vaccines, both past and present, whilst also discussing both the development of various vaccine adjuvants and their advantages in vaccination strategies against influenza infection. The manuscript is well-written and provides the reader with an interesting introduction to vaccine adjuvants. The manuscript could do with some clarifications in the text to improve it, which I will highlight in my separate notes for each section of the manuscript.
The manuscript draws together a lot of knowledge from papers published over the years on the development of vaccine adjuvants and the successes in including those adjuvants in influenza vaccines. The authors successfully demonstrate the need to move to so-called improved influenza vaccines, beyond the various non-adjuvanted influenza vaccines commonly used in seasonal influenza vaccination strategies, given that some members of the target populations do not respond adequately to vaccination to provide sufficient protection against infection.
For a general comment not limited to one section of the manuscript, the authors need to be clear to the reader that they explain what they mean by vaccine efficacy or effectiveness, i.e. efficacy or effectiveness against what outcome (infection, hospitalisation, mortality, etc.). Discussing efficacy or effectiveness in very general terms without referring to the clinical outcome is not helpful for the reader.
COMMENTS ON THE INTRODUCTION SECTION
The research question is clearly stated, the authors sought to discuss the challenges in the development of adjuvants for influenza vaccines and to provide the reader with justifications for their use.
Line 29: “...epidemics are caused every winter by seasonal influenza viruses (i.e., A/H1N1…” should read “epidemics are caused every winter by several different circulating strains of seasonal influenza viruses (e.g., A/H1N1…”. The authors should be careful with the notation used to denote the different strains, as some times in the manuscript they use A/H1N1 whereas others they use A(H1N1), etc. The authors ought to check how each strain is written for consistency. I’m not aware of a preferred approach in the literature, just as long as the choice by the authors is followed throughout the manuscript.
Line 42: “...as of April 24, 2023” should read “...as of April 24, 2023, with limited evidence of sustained human-to-human transmission.”
Line 44: “...the effectiveness of current influenza vaccines is suboptimal” should read “...the effectiveness of current influenza against infection can be suboptimal”, and the authors ought to include a brief summary of the reasons why effectiveness can be suboptimal. The authors have mentioned several reasons in the manuscript but I think it’s better to have those reasons listed upfront. The authors should at least mention the fact that seasonal influenza vaccines are manufactured several months before an influenza season begins and the vaccines contain only 3 or 4 strains of the virus, which may not be the dominant circulating strains several months later, and/or those circulating strains may have antigenically drifted or shifted when the season begins.
COMMENTS ON THE 'CURRENT LICENSED INFLUENZA VACCINES' SECTION
Line 51: “Annual vaccination is one of the most effective ways to prevent seasonal influenza” should read “Annual vaccination is the most effective way to prevent seasonal influenza infection”
Line 95: “The nasal spray vaccine contains A/H1N1 and A/H3N2…” should read “The nasal spray vaccine used in recent seasonal influenza seasons contains…” then the influenza strains mentioned should follow the format decided by the authors, i.e. A/H1N1 or A(H1N1).
Line 119-120: “The 2021-2022 VE against A(H3N2)...” may need some clarifications. The authors are not incorrect in their writing here, it’s just not clear why they’re mentioning the VE against A/H3N2 specifically. Was A/H3N2 the dominant circulating strain and VE was particularly low in that season? Were estimates available against other circulating strains?
Line 123-127: “Initially, VE was reported…to clinical sites.” I don’t see the relevance of this sentence. This sentence could be removed from the manuscript as it doesn’t add anything to the arguments made by the authors.
COMMENTS ON THE 'ADJUVANTS USED FOR LICENSED INFLUENZA VACCINES' SECTION
Line 180-181: “People aged 65 years…of flu-related hospitalisations” doesn’t follow from what was said in the sentence before. This information is interesting and useful but it should be mentioned far earlier in the manuscript, probably in the section devoted to setting the epidemiological context for influenza vaccination and describing the epidemiological burden of influenza.
Line 201-202: “Narcolepsy was reported more frequently than usual in clinical trials of vaccines containing AS03” needs some additional information and context. If we are to consider the pandemic influenza vaccine made by GSK in 2009 then I believe the literature is mixed when it comes to the trigger for the narcolepsy in vaccinated recipients, with newer publications pointing more towards other ingredients of the vaccine being the likely trigger rather than the adjuvant. The authors should be absolutely clear on this situation in their manuscript, so more context is needed, ideally with as much information as is possible to present on the agreed trigger for narcolepsy.
COMMENTS ON THE 'ADVERSE EFFECTS RELATED TI ADJUVANTED VACCINES' SECTION
Line 225-226: “AS03-adjuvanted…in some countries” needs to be addressed. As above, the agreed cause of narcolepsy in vaccinated individuals is, to my knowledge having consulted this issue recently, not the adjuvant but other ingredients in the vaccine. If this is indeed the case then it is important that this is clarified in the text otherwise the implication is that narcolepsy was caused by the adjuvant.
COMMENTS ON THE 'PROGRESS IN THE DEVELOPMENT OF ADJUVANTS FOR INFLUENZA VACCINES' SECTION
No comments.
COMMENTS ON THE 'CONCLUSIONS' SECTION
No comments.
COMMENTS ON THE TITLE AND ABSTRACT
The title accurately reflects the work done for the manuscript.
The abstract contains sufficient information to describe the manuscript, although some changes need to be made:
Line 12: “effectiveness is suboptimal” should read “effectiveness against infection can be suboptimal”.
Line 12: “is not sufficient” should read “can be insufficient”.
Line 12-13: “vaccine efficacy is lower” should read “vaccine efficacy against infection is usually lower”.
Line 14-15: “licensed in the United States” should ready “licensed in the United States and other countries”.
Line 20: “but are also effective” should read “but can also be effective”.
Line 21: “In this review” should read “In this narrative review” so that the reader understands what sort of review has been written.
Line 21: “we provide an overview of adjuvants influenza vaccines and discuss their future.” undersells the content of the manuscript. I would recommend “we provide an overview of influenza vaccines, both past and present, before presenting a discussion of adjuvanted influenza vaccines and their future.”
Author Response
[Reviewer#3]:
The manuscript submitted by Shichinohe and Watanabe presents a narrative review of influenza vaccines, both past and present, whilst also discussing both the development of various vaccine adjuvants and their advantages in vaccination strategies against influenza infection. The manuscript is well-written and provides the reader with an interesting introduction to vaccine adjuvants. The manuscript could do with some clarifications in the text to improve it, which I will highlight in my separate notes for each section of the manuscript. The manuscript draws together a lot of knowledge from papers published over the years on the development of vaccine adjuvants and the successes in including those adjuvants in influenza vaccines. The authors successfully demonstrate the need to move to so-called improved influenza vaccines, beyond the various non-adjuvanted influenza vaccines commonly used in seasonal influenza vaccination strategies, given that some members of the target populations do not respond adequately to vaccination to provide sufficient protection against infection.
For a general comment not limited to one section of the manuscript, the authors need to be clear to the reader that they explain what they mean by vaccine efficacy or effectiveness, i.e. efficacy or effectiveness against what outcome (infection, hospitalisation, mortality, etc.). Discussing efficacy or effectiveness in very general terms without referring to the clinical outcome is not helpful for the reader.
In response to the reviewer’s comment, we have added either ‘infection’ or ‘hospitalization’ after ‘vaccine efficacy’ or ‘vaccine effectiveness’ in the revised manuscript (page 1, line 13; page 2, line 52; page 2, line 65-66; page 3, line 144; page 4, line 153; page 6, line 162; page 10, line 344).
COMMENTS ON THE INTRODUCTION SECTION
The research question is clearly stated, the authors sought to discuss the challenges in the development of adjuvants for influenza vaccines and to provide the reader with justifications for their use.
Line 29: “...epidemics are caused every winter by seasonal influenza viruses (i.e., A/H1N1…” should read “epidemics are caused every winter by several different circulating strains of seasonal influenza viruses (e.g., A/H1N1…”. The authors should be careful with the notation used to denote the different strains, as some times in the manuscript they use A/H1N1 whereas others they use A(H1N1), etc. The authors ought to check how each strain is written for consistency. I’m not aware of a preferred approach in the literature, just as long as the choice by the authors is followed throughout the manuscript.
In response to the reviewer’s comment, we have modified the sentence (page 1, lines 29-31) and standardized the terminology used to indicate the subtypes (e.g., A/H1N1 or A/H3N2) throughout the revised manuscript (page 3, line 116; page 3, line 140).
Line 42: “...as of April 24, 2023” should read “...as of April 24, 2023, with limited evidence of sustained human-to-human transmission.”
We have modified the statement in the revised manuscript (page 1, line 45).
Line 44: “...the effectiveness of current influenza vaccines is suboptimal” should read “...the effectiveness of current influenza against infection can be suboptimal”,
We have modified the sentence in the revised manuscript (page 2, line 47).
…and the authors ought to include a brief summary of the reasons why effectiveness can be suboptimal. The authors have mentioned several reasons in the manuscript but I think it’s better to have those reasons listed upfront. The authors should at least mention the fact that seasonal influenza vaccines are manufactured several months before an influenza season begins and the vaccines contain only 3 or 4 strains of the virus, which may not be the dominant circulating strains several months later, and/or those circulating strains may have antigenically drifted or shifted when the season begins.
In response to the reviewer’s comment, we have added the reasons why the current seasonal influenza vaccine is suboptimal “upfront” (page 2, line 48-51).
COMMENTS ON THE 'CURRENT LICENSED INFLUENZA VACCINES' SECTION
Line 51: “Annual vaccination is one of the most effective ways to prevent seasonal influenza” should read “Annual vaccination is the most effective way to prevent seasonal influenza infection”
We have modified the sentence in the revised manuscript (page 2, line 58).
Line 95: “The nasal spray vaccine contains A/H1N1 and A/H3N2…” should read “The nasal spray vaccine used in recent seasonal influenza seasons contains…” then the influenza strains mentioned should follow the format decided by the authors, i.e. A/H1N1 or A(H1N1).
We have modified the sentence in the revised manuscript (page 3, line 114-115).
Line 119-120: “The 2021-2022 VE against A(H3N2)...” may need some clarifications. The authors are not incorrect in their writing here, it’s just not clear why they’re mentioning the VE against A/H3N2 specifically. Was A/H3N2 the dominant circulating strain and VE was particularly low in that season? Were estimates available against other circulating strains?
We discussed only the 2021-2022 VE against A/H3N2 because most influenza viruses detected in the US during the 2021-2022 influenza season were A/H3N2 viruses. However, A/H1N1 VE is also a concern; therefore, we have changed the reference to the reported European data (Kissling et al., Influenza Other Respi Viruses, 2023) (page 3, line 138-141).
Line 123-127: “Initially, VE was reported…to clinical sites.” I don’t see the relevance of this sentence. This sentence could be removed from the manuscript as it doesn’t add anything to the arguments made by the authors.
We have deleted this sentence as suggested (page 3, line 141).
COMMENTS ON THE 'ADJUVANTS USED FOR LICENSED INFLUENZA VACCINES' SECTION
Line 180-181: “People aged 65 years…of flu-related hospitalisations” doesn’t follow from what was said in the sentence before. This information is interesting and useful but it should be mentioned far earlier in the manuscript, probably in the section devoted to setting the epidemiological context for influenza vaccination and describing the epidemiological burden of influenza.
In response to the reviewer’s comment, we have moved the sentences, “people 65 years and older account for 70%-85% of flu-related deaths and 50%-70% of flu-related hospitalizations” and “this population generally shows lower immune responses and vaccine effectiveness” to the Introduction section of the revised manuscript (page 1, lines 34-35 and page 2, lines 51-52, respectively).
Line 201-202: “Narcolepsy was reported more frequently than usual in clinical trials of vaccines containing AS03” needs some additional information and context. If we are to consider the pandemic influenza vaccine made by GSK in 2009 then I believe the literature is mixed when it comes to the trigger for the narcolepsy in vaccinated recipients, with newer publications pointing more towards other ingredients of the vaccine being the likely trigger rather than the adjuvant. The authors should be absolutely clear on this situation in their manuscript, so more context is needed, ideally with as much information as is possible to present on the agreed trigger for narcolepsy.
In response to the reviewer’s comment, we now describe how recent studies suggest that the trigger for narcolepsy may be viral antigen(s) of the pandemic H1N1 influenza viruses, not AS03 (page 8, lines 253-259).
COMMENTS ON THE 'ADVERSE EFFECTS RELATED TI ADJUVANTED VACCINES' SECTION
Line 225-226: “AS03-adjuvanted…in some countries” needs to be addressed. As above, the agreed cause of narcolepsy in vaccinated individuals is, to my knowledge having consulted this issue recently, not the adjuvant but other ingredients in the vaccine. If this is indeed the case then it is important that this is clarified in the text otherwise the implication is that narcolepsy was caused by the adjuvant.
As the reviewer pointed out, the recent studies suggest that the local CD4+ T-cell response recognizing the hypocretin peptide may have been stimulated by the viral antigen(s) of the pandemic H1N1 influenza viruses due to the molecular mimicry between hypocretin and viral antigen peptides, triggering a cross-reactive autoimmune response targeting hypocretin neurons. We have clarified this point in the text (page 8, line 253-259).
COMMENTS ON THE TITLE AND ABSTRACT
The title accurately reflects the work done for the manuscript.
The abstract contains sufficient information to describe the manuscript, although some changes need to be made:
Line 12: “effectiveness is suboptimal” should read “effectiveness against infection can be suboptimal”.
In response to the reviewer’s comment, we have modified the sentence in the revised manuscript (page 1, lines 11-12).
Line 12: “is not sufficient” should read “can be insufficient”.
In response to the reviewer’s comment, we have modified the sentence in the revised manuscript (page 1, line 12-13).
Line 12-13: “vaccine efficacy is lower” should read “vaccine efficacy against infection is usually lower”.
In response to the reviewer’s comment, we have modified the sentence in the revised manuscript (page 1, lines 12-13).
Line 14-15: “licensed in the United States” should ready “licensed in the United States and other countries”.
In response to the reviewer’s comment, we have modified the sentence in the revised manuscript (page 1, lines 14-15).
Line 20: “but are also effective” should read “but can also be effective”.
In response to the reviewer’s comment, we have modified the sentence in the revised manuscript (page 1, lines 20-22).
Line 21: “In this review” should read “In this narrative review” so that the reader understands what sort of review has been written.
In response to the reviewer’s comment, we have modified the sentence in the revised manuscript (page 1, line 22).
Line 21: “we provide an overview of adjuvants influenza vaccines and discuss their future.” Undersells the content of the manuscript. I would recommend “we provide an overview of influenza vaccines, both past and present, before presenting a discussion of adjuvanted influenza vaccines and their future.”
In response to the reviewer’s comment, we have modified the sentence in the revised manuscript (page 1, lines 22-23).

Round 2
Reviewer 2 Report
The authors have satisfactorily responded to my concerns.